# *SHOX* Deletion and Idiopathic Short Stature: What Does the Clinician Need to Know? Case Series Report

**DOI:** 10.3390/diagnostics13010105

**Published:** 2022-12-29

**Authors:** Maria-Christina Ungureanu, Anamaria Hrisca, Lavinia Caba, Laura Teodoriu, Stefana Bilha, Cristina Preda, Letitia Leustean

**Affiliations:** 1Endocrinology Department, “Grigore T. Popa” University of Medicine and Pharmacy, 700111 Iasi, Romania; 2Medical Genetics Department, “Grigore T. Popa” University of Medicine and Pharmacy, 700111 Iasi, Romania

**Keywords:** hypostature, subclinical skeletal dysplasia, *SHOX* mutations

## Abstract

Children diagnosticated with idiopathic short stature (ISS) are probably, in most cases, underdiagnosticated. The genetic causes of ISS may be mutations of genes involved in local regulation of the growth plate or genes involved in the GH-IGF1 axis physiology. We present a kindred of five children evaluated for short stature or low normal stature, initially diagnosticated as idiopathic short stature, familial short stature, or being small for gestational age. Clinical signs suggestive of *SHOX* deletion screening in a child with short stature are low arm span/height ratio, increased sitting height/height ratio, BMI > 50% percentile, Madelung deformity, cubitus valgus, bowing and shortening of the forearm, dislocation of the ulna (at the elbow), and the appearance of muscular hypertrophy. Radiological characteristics suggestive of *SHOX* deficiency are triangularisation of the distal radial epiphysis, an enlarged diaphysis of the radius plus bowing of the radius, the convexity of the distal radial metaphysis, short fourth and fifth metacarpals, pyramidalization of the carpal row. Treatment with rGH is approved for children with *SHOX* gene deficiency and short stature. This kindred is an example that familial short stature, idiopathic short stature, and short stature due to a small gestational age are not final diagnoses. Complex investigations are necessary to identify the precise cause, leading to optimal clinical management. Treatment with rGH is an option for some of them; for others, it has no therapeutic response and, in some cases, is even harmful.

## 1. Introduction

One of the most common pathologies investigated by a pediatric endocrinologist is short stature. Adult body height depends substantially on the length of the long bones. Longitudinal growth is a complex developmental process influenced by multiple environmental and genetic factors [1]. The causes of stature deficit may usually be established by a careful medical history, clinical examination, hormonal, and imaging exploration. In addition, in recent decades, tremendous progress has been made in identifying monogenic causes of growth disorders [2,3].

Idiopathic short stature (ISS) is usually defined in children as GH-sufficient (normal response at provocative tests), with height < −2 SD for age, sex, and population group, in the absence of chromosomal, systemic, endocrine, or nutritional abnormalities, normal birth weight and length (birth weight and length > −2 SD), harmonious small stature and absence of major dysmorphic features, without psychosocial problems, and with proper nutrition [3]. Idiopathic short stature or small for gestational age (SGA) are not definitive diagnoses. Labelling children only as having idiopathic low stature or SGA after initial clinical, endocrine, and radiological evaluation explains the highly variable response to growth hormone treatment in these cases. In some rare situations, growth hormone treatment can be harmful, as in Bloom syndrome or Nijmegen syndrome.

Since short stature is a common clinical presentation, it is widely accepted that it is caused by variants with polygenic inheritance. Recent studies have proposed that many children classified as ISS could have a monogenic defect with the mildest spectrum of syndromic conditions. In these cases, growth impairment may be caused by defects of the GHRH-GH-IGF1 cascade or factors involved in growth plate development. The latest are gene mutations of different local factors associated with short stature and phenotypes varying from severe, disharmonious dwarfism to subclinical skeletal dysplasia [2,4,5].

## 2. Materials and Methods

We present a kindred of five children (three boys and two girls), one of the boys having another father, raised in different foster homes, with their mother having short stature (145 cm, meaning −3.42 SD). For clinical evaluation, growth charts for the Romanian population were used [6]. This study was approved by the local Ethical Committee; informed consent was signed by the guardians; and they approved the publication of the medical data of the children.

### Genetic Testing

DNA was extracted from peripheral blood samples stored with an EDTA agent using Wizard Genomic DNA Purification Kit (Promega Corp., Madison, WI, USA) or Invitrogen PureLink^®^ Genomic DNA Kit (Invitrogen, Waltham, MA, USA). The MLPA analysis was performed according to the manufacturer’s protocol. We used a commercially available P018 SALSA MLPA kit (MRC-Holland, Amsterdam, The Netherlands). Data analysis was performed using the Coffalyser.NET software (MRC-Holland, Amsterdam, The Netherlands).

## 3. Results

### 3.1. Case 1

The first case (Figure 1) is a thirteen-year-old boy, a full-term birth weighing 2800 g. He is a stepbrother in the other four cases. The first evaluation, at the age of eight, showed short stature (H = 117 cm, −2.56 SD), a BMI of 19 Kg/m^2^ (56th percentile), and a mid-parental height (MPH) of 163.1 cm (−2.14 SD). The clinical exam revealed an abnormal development with mezomelic short stature: height 137 cm (−2.64 SD), a BMI of 20.1 Kg/m^2^ (77th percentile), puberty P3G3, arm span = 136 cm, sitting height = 76 cm, subischial leg length = 62 cm, extremities/trunk ratio = 2.6 (<2.64), sitting height/height = 0.55 (>2.5 SD). The hormonal assessment revealed normal thyroid status, GH sufficient. Finally, the genetic test showed the *SHOX* gene mutation: heterozygous deletion of approximately 1.21 MB in the Xp22PAR region between genomic positions 500.427 and 1.712.090 (hg19), using the MLPA technique.

### 3.2. Case 2

The second case (Figure 2) is a nine-year-old boy born small for gestational age (SGA): weight 2400 g and length 46 cm at 40 weeks of pregnancy (−2.45 SD). At the age of four, the physical examination revealed harmonious short stature: height = 94 cm (−3.6 SD) and weight = 12 kg and a BMI in the 2nd percentile. All hormonal parameters were within normal limits. Considering the diagnosis of SGA, treatment with rhGH was initiated at a dose of 0.04 mg/day. After five years of treatment, the height is 123 cm (−2.66 DS) and the extremities-to-trunk ratio was 2.63, excluding a mezomelic short stature at this moment. Considering the kinship with Case No. 1, we have performed a genetic test that confirms a *SHOX* gene mutation: a heterozygous deletion of approximately 1.21 MB in the Xp22PAR region between genomic positions 500.427 and 1.712.090 (hg19).

### 3.3. Case 3

The third case (Figure 3) is an eight-and-a-half-year-old boy, born prematurely with 1700 g at 34 weeks of gestation (>−2 SD). He was diagnosed at seven and a half with short stature: height 110 cm (−3.2 SD), weight 16 kg, BMI 3rd percentile, and MPH 170.4 cm (−1.03 SD). The genetic test confirms the *SHOX* gene mutation. Treatment with rGH was initiated. At 12 months of treatment, the height was 118 cm (−2.68 SD). No abnormal development with mezomelic short stature was noted: extremities-to-trunk ratio = 2.58 (>2.54) and sitting height-to-height ratio = 0.55 (>2.5 SD). On the hand X-ray, shortening of the fourth metacarpal bone and discrete convexity of the distal radial metaphysis was noted. The genetic test confirms the *SHOX* gene mutation: heterozygous deletion of approximately 1.21 MB in the Xp22PAR region between genomic positions 500.427 and 1.712.090 (hg19).

### 3.4. Case 4

The fourth case (Figure 4) is a five-year-old girl with a low normal birth weight (2300 g at 36 weeks). Clinical exam revealed: height = 103 cm (−1.6 SD), weight = 16.5 kg, BMI = 29th percentile, and MPH = 170.4 cm (−1.03 SD). She tested positive for *SHOX* deletion due to her family history—a heterozygous deletion of approximately 1.21 MB in the Xp22PAR region between genomic positions 500.427 and 1.712.090 (hg19). Treatment with rGH was postponed due to our country’s guidelines for GH treatment for short stature.

### 3.5. Case 5

The fifth case (Figure 5) is an 11-year-old girl without short stature: height 141 cm (−1.41 SD), weight 32.5 kg, BMI 16 kg/m^2^, and MPH 170.4 cm (−1.03 SD). She tested negative for *SHOX* deletion

## 4. Discussion

The longitudinal bone growth is a coordinated process of chondrocyte proliferation, maturation, and hypertrophy; matrix synthesis with vascular invasion at the end; and migration of osteoblasts/osteocytes, and other bone marrow cell types. Critical regulators of cartilage and bone formation are:-Hormones: GH, IGFs, thyroid hormone, vitamin D, glucocorticoids, oestrogens, and androgens;-Paracrine factors: retinoic acid, Indian hedgehog protein, and PTHrP (PTH related peptide), BMPs (bone morphogenetic proteins), FGFs (fibroblast growth factors), and their receptors, CNP (C-type natriuretic peptide), and its receptor;-Proinflammatory cytokines—TNF, IL-1β, and IL-6;-Extracellular matrix molecules and intracellular proteins.

Anomalies of local genetic factors may cause a relatively harmonious short stature— “subclinical skeletal dysplasia” [1,4].

Fibroblast growth factor receptor 3 (FGFR3) is a crucial negative regulator in chondrocyte proliferation: it reduces the proliferation activity of chondrocytes and their differentiation speed and decreases cartilage matrix synthesis. Activating mutations in *FGFR3* cause thanatophoric dysplasia, achondroplasia, and hypochondroplasia, but, recently, alterations in the FGFR3 gene have been associated with proportionate short stature [7].

C-type natriuretic peptide (PNC), encoded by the gene *NPPC*, and its receptor, natriuretic peptide receptor (NPR2), encoded by the gene *NPR2*, are expressed mainly in the brain, where it is thought to act as a neuropeptide, and in the hypertrophic area of the growth plate. They have a role in chondrocyte proliferation and differentiation, cartilage matrix synthesis, and inhibiting FGFR3 activity [5,8,9]. Inactive mutations of the CNP receptor gene (*NPR2*) cause acromesomelic dysplasia type Maroteaux (homozygous mutations). *NPR2* gene and *NPPC* gene heterozygous mutations have been identified in patients with idiopathic hypostature, abnormal body proportion, and nonspecific skeletal abnormalities [9,10].

Indian Hedgehog protein (IHH) is essential in the proliferation and differentiation of chondrocytes through the PTHRP level and activation of the PP receptor (PTH/PTHrP receptor). Homozygous mutations of IHH cause acro-capito-femoral dysplasia. Heterozygous mutations cause short stature with variable brachydactyly (type A1-BDA1) and craniosynostosis. Growth hormone treatment may improve the final height of these children [5,11].

Aggrecan, encoded by the *ACAN* gene, is the most abundant proteoglycan in hyaline cartilage. It plays a pivotal role in cartilage formation, bone morphogenesis, and the biomechanical properties of cartilage [3,5,12]. Homozygous mutations cause spondylo-epi-metaphyseal dysplasia. Haploinsufficiency of the ACAN gene has been reported in patients with (a) idiopathic, harmonious short stature but with advanced bone age, (b) disharmonic short stature with brachydactyly, craniofacial dysmorphia, or in children with intrauterine growth delay and an early installation of osteoarthritis [9,13,14].

Short Stature Homeobox (SHOX) Protein, encoded by *SHOX* gene.

The SHOX protein contains a homeodomain, a structure frequently seen in transcription factors involved in body patterning. As a transcription activator, it stimulates and coordinates the proliferation and differentiation of chondrocytes, increases natriuretic peptide B (NPPB) transcription, inhibits *FGFR3* gene expression, and interacts with the SOX trio (*SOX9, SOX5*, and *SOX6* genes), which function as the significant chondrogenic factors activating the enhancer of the *ACAN* gene [1,9,15,16]. *SHOX* haploinsufficiency has become the main recognized monogenic cause, responsible for 2.6% of the nonsyndromic cases of short stature [3,17]. The *SHOX* gene is located on the tip of the short arms of both sex chromosomes X and Y, inside the telomeric part of pseudoautosomal region 1 (PAR1) of the X chromosome at Xp22.3 and the Y chromosome at Yp11.3. Genes in this region of the sex chromosomes do not undergo X-inactivation; therefore, two active copies are available for normal skeletal growth and development in both males (46, XY) and females (46, XX) [17,18]. There is no difference between the *SHOX* gene in female and male subjects. The *SHOX* region is a hotspot for frequent recombination events between the X and Y chromosomes [18]. A spatiotemporally restricted expression pattern of *SHOX* exists in the middle portion of the arm, elbow, radius, ulna, and a few wrist bones; on the lower limbs, the expression resembles the upper limb development. *SHOX* is also present in the first and second pharyngeal arches (development of the maxilla, mandible, and some ear bones). Another gene—*SHOX2*—is also expressed in limbs and the developing heart [1,15,16].

### 4.1. Molecular Basis of SHOX Haploinsufficiency

*SHOX* has seven exons encoding amino acid sequences (excluding exon 1), five introns, three untranslated regions, and eleven regulatory regions, all of which are necessary for correct gene expression [16].

*SHOX* gene function is dosage-dependent, and the phenotypic aspects range from very short, dysmorphic stature (Langer syndrome) to Leri–Weill dyschondrosteosis (LWD) and idiopathic short stature, respectively. *SHOX* haploinsufficiency frequently results from intragenic mutations or deletions and copy number variations (CNVs) in the gene-flanking regions (PAR1 region involving *SHOX* exons and/or the *cis*-acting enhancers) [1,18,19]. Additionally, 80% of all the described mutations are gene deletions of different sizes, encompassing the SHOX gene itself or a regulatory enhancer region located 50–250 kb downstream of the coding region. For 20% of mutations, they are missense and nonsense mutations in exons 3 and 4. They encode the important functional homeodomain or microdeletions encompassing only one or more exons. These mutations are predicted to cause the protein’s inactivation or block nuclear translocation or dimerization of *SHOX* [20]. These chromosome rearrangements disrupt the interaction between the enhancer regions and the promoter of *SHOX*, resulting in decreased transactivation of gene expression. Particular attention should be paid to the regulatory regions of the SHOX gene, more commonly associated with isolated low stature. In contrast, exonic point mutations account for a small percentage of cases [15].

SHOX homozygous mutations or compound heterozygous mutations cause Langer syndrome, a very rare condition with severely disproportionate short stature (mesomelic dysplasia due to the underdeveloped or absent ulna and fibula) and the Madelung deformity. *SHOX* haploinsufficiency underlies the short stature of Turner syndrome patients and is associated with ISS and LWD. It is involved in 2–3% of ISS cases and ∼70% of LWD cases [15]. LWD is a mesomelic short stature, moderate/mild skeletal dysplasia with Madelung deformity, highly arched palate, cubitus valgus, and scoliosis.

The short stature associated with *SHOX* gene deletion ranges from 135 cm to normal height. The reason for this variability is not known [1]. By definition, short stature is present in 3% of the population; if we consider SHOX defects responsible for 3% of the nonsyndromic cases of short stature, we should expect a population prevalence of 1 in 1000 [21].

The signs are more frequent and more severe in girls. It may be explained by higher estrogen levels, by searching for such signs (common with Turner Syndrome signs) more often in girls with short stature, or because the SHOX on the X is more prone to getting deleted than the Y. The phenotype usually becomes more pronounced with age, and typical manifestations appear over time [1,15,22]. The mean adult height of ISS patients with *SHOX* haploinsufficiency and the normal karyotype is around –2.2 SD (a mean growth deficit of about 12 cm), but it may be more severe in patients with LWD phenotypes. In Turner syndrome, the height deficit is about –3.2 SD; possibly in these cases, the short stature is not explained only by SHOX haploinsufficiency [15].

Overdosage of the *SHOX* gene by large duplications of *SHOX* exons and *cis*-acting enhancers is implicated in the tall stature of Klinefelter syndrome (47, XXY) and triple-X syndrome (47, XXX). Still, small duplications may reduce *SHOX* expression levels by disrupting the *cis*-regulatory machinery causing low stature [15].

Clinical signs that are more or less specific for *SHOX* deletion:-Low arm span/height ratio;-Height sitting height/height ratio;-Extremity-to-trunk ratio < 1.95 + 0.5× height (metric).

The arm span is significantly reduced compared to the standing height. The leg length is considerably shorter than expected from the sitting height due to the mesomelia associated with *SHOX* deficiency. Binder introduced the extremities-to-trunk ratio, a formula that integrates the three parameters. The ratio is the sum of the leg length and arm span divided by the height of the trunk (the sitting height). It is not applicable to children younger than six years old when this body disproportion is normal. Longitudinal follow-up studies of female patients with *SHOX* haploinsufficiency showed that body disproportion often deteriorates during puberty. Hence, a normal ratio in a schoolgirl excludes *SHOX* deficiency with a high negative predictive value of almost 100% [15,23].

-BMI > 50th percentile

BMI above the 50th percentile for age/sex is a significant predictor, reflecting disproportion due to decreased leg length and muscular hypertrophy rather than increased fat mass. Notably, children with idiopathic short stature without *SHOX* deficiency typically have a decreased BMI [24].

-Muscular hypertrophy appearance, especially in the calves, has been reported in about one-third of *SHOX*-deficient patients [24].-Madelung deformity

Madelung deformity consists of the bilateral forearm’s bones (the radius and the ulna) bowing and shortening with dislocation of the ulna. Histopathological analysis showed a disturbed columnar arrangement of chondrocytes in the growth plate and abnormal endochondral ossification with premature fusion of the distal radial epiphysis [15]. Madelung deformity consists of bowing and shortening the radius, prominence of the ulnar head, palmar and ulnar deviation of the carpal bones, and dorsal subluxation of the ulnar head. The ulna protrudes by partial dislocation in the fist, elbow, or both. The distal radial epiphysis fusions are premature, probably due to an aberrant cell death process in the growth plate and an aberrant ligament that binds the lunate to the radius. This ligament compresses the epiphysis of the radius and disturbs linear bone growth. Madelung deformity is a characteristic feature of LWD but may also occur in other disorders such as epiphyseal dysplasia, mucopolysaccharidosis, pseudo-hypoparathyroidism type 1b, and injury [1,15,21].

-Cubitus valgus with increased carrying angle of the elbow;-Bowing and shortening of the forearm;-Shortening of the fourth and fifth metacarpals;-High arched palate;-Scoliosis;-Micrognathia [15,22,25].

Rappold et al. developed a scoring system that includes the body segment ratio and other indices of SHOX deficiency. At a cutoff score of 7 (of a total score of 24), the positive prediction rate for identifying a *SHOX* gene point mutation or deletion was 19%. However, downstream or upstream SHOX enhancers are unknown at the moment [25]. In addition, another diagnostic algorithm was elaborated in 2020 by Vannelli et al. [26].

Phenotypic variation exists among individuals with SHOX deficiency because a high-arched palate is more prevalent in Turner syndrome, whereas Madelung deformity, a short forearm and lower leg, bowing of the forearm and tibia, and muscular hypertrophy are more prevalent in Léri–Weill syndrome [17].

In Turner syndrome, there are probably other factors (haploinsufficiency of other genes on the X chromosome, chromosomal imbalance) that contribute to the short stature. The SHOX deficiency in Turner syndrome may explain cubitus valgus, Madelung deformity, mesomelia, disproportionate skeletal sizes, a high-arched palate, and micrognathia [17].

Radiological characteristics in SHOX deficiency are [26] (Figure 6):o Triangulation of the distal radial epiphysis;o Lucency of the ulnar border of the distal radius;o The enlarged diaphysis of the radius and bowing of the radius;o Short fourth and fifth metacarpal;o Pyramidalization of the carpal row;o Convexity of the distal radial metaphysis.

SHOX-deficient short stature should be suspected in children with a first-degree relative with SHOX deficiency and one of the following (Figure 7) [18,21,25,26,27,28]:-Disproportionate short stature (young school age);-Madelung deformity (older school age);-Short stature and specific minor abnormalities (see Clinical Characteristics, Clinical Description).

Several testing algorithms have been proposed for children with idiopathic short stature:o By Binders et al. based on clinical, auxologic, and radiologic criteria [15,21];o By Vannelli et al. [26].

The phenotype is not correlated with the size and type (missense or frameshift) of mutations, with identical *SHOX* abnormalities being detected in patients with ISS and LWD individuals with normal stature. Hormonal status seems to influence the phenotype expression of *SHOX* haploinsufficiency (more severe in postpubertal female patients). The mutations involving only the downstream enhancer regions lead to slightly milder phenotypes with more harmonious short stature than mutations or deletions in the exons [15,18].

### 4.2. Molecular Analysis

The first approach for molecular diagnosis of *SHOX* haploinsufficiency should be multiplex ligation-dependent probe amplification (MLPA). The assay allows multiple targets to be amplified with a single pair of primers, has high sensitivity, detects and quantifies small deletions in multiple samples relatively quickly (superior to FISH), and with exons and marginal regions sequencing. A kit, P018 SALSA MLPA (MRC-Holland bv; Willem Schoutenstraat 1 1057 DL, Amsterdam, the Netherlands), was used for *SHOX* analysis in the presented cases.

### 4.3. Short Stature Treatment in SHOX Deficiency

rhGH therapy for short stature due to *SHOX* deficiency is approved. There are no concerns regarding treatment safety [29]. In a prospective, open-label, randomized study, including patients with *SHOX* deficiency treated with recombinant GH, the final adult height was more than −2 SD in 57% of the patients [30]. GH treatment in subjects with varying forms of *SHOX* deficiency increases height velocity. Height gains from treatment initiation to adult height were about 1.1–1.2 SDS (∼7 cm)—as cited in Turner syndrome. The treatment response is better in patients with deletions in the *SHOX* gene enhancer area than in those with deletions in the *SHOX* gene. The maxim catch-up growth occurs in the first 12 months of treatment.

There are also psychosocial factors such as self-image, social integration, and school performance that may be influenced by height [17]. Adverse events in GH treatment patients with *SHOX* are like those reported in other GH-treated paediatric populations [31]. However, the period of study for GH administration is relatively short. Therefore, long-term follow-up of a significant number of patients treated with GH for *SHOX* deficiency is required. There is no systematic effect on the skeletal anomalies of this disorder induced by growth hormone treatment. The combination therapy of GH and GnRHa may effectively promote stature growth in patients with *SHOX* deficiency by delaying premature fusion of the growth plates. Still, the clinical experience with combined therapy is poor. The combination therapy of GH and aromatase inhibitors for decreasing bone age maturation cannot be applied to girls. The long-term safety and efficacity are still uncertain [1].

In our kindred, Case no. 1 had typical clinical aspects for SHOX deletion: low height, BMI > percentile 50%, appearance of muscular hypertrophy, mezomelic short stature with extremities/trunk ratio < 2.64 and sitting height/height > 2.5 SD. Children with ISS and a low mean parental height do not exclude rGH treatment because it may be *SHOX* haploinsufficiency. The risk of a child inheriting the SHOX gene deficiency is 50% when one of the parents is affected; if both parents are SHOX gene-deficient, there is a 50% risk of moderate-to-mild hypostature by SHOX deficiency, a 25% chance of severe Langer dwarfism, and a 25% chance of having none of the conditions [26].

Case no. 2 has SGA, a low BMI, and no clinical aspects suggestive of *SHOX* deletion, but we may explain the modest rGH treatment response with the genetic diagnostic. He has a first-degree relative with *SHOX* deficiency. Case no. 3 has short stature, a low BMI, a normal birth weight for gestational age, and discrete radiological signs of skeletal dysplasia; he was tested positive for *SHOX* deletion. Children with SGA who do not have catch-up growth until the age of four probably have an unspecified growth disorder with prenatal onset, and the cause is generally as “idiopathic” as the ISS. Cases 4 and 5 do not have short stature (height > −2 SD).

Genetic testing is always recommended for short stature if it is severe or associated with microcephaly/relative macrocephaly, dysmorphic traits, disproportions of body segments highlighted by auxological determinations, intellectual disability, and a positive family history [3,32].

## 5. Conclusions

The nonspecific and variable phenotype of SHOX deficiency frequently leads to diagnostics such as skeletal dysplasia with disharmonious short stature, ISS, or SGA. Rapaport et al. stated in their article [3] that short children born SGA should be classified in the same group as ISS. Complex investigations are needed to identify precise aetiology, leading to optimal clinical management. The genetic assessment also permits to exclude conditions associated with short stature for which rhGH treatment is contraindicated, such as chromosomal instability and DNA repair defects syndromes due to the risk of neoplasia (e.g., Bloom syndrome, Nijmegen syndrome), debatable (e.g., neurofibromatosis type1), or where higher doses are needed for an appropriate growth response (e.g., IGF1R haploinsufficiency, SHOX haploinsufficiency). A mutation in FGFR3 may cause, rarely, proportionate short stature, especially in families with an autosomal dominant inheritance pattern. A C-type natriuretic peptide analogue (Vosoritide) was developed to treat short stature from achondroplasia. CNP is a potent stimulator of endochondral ossification and could counteract the negative signal of FGFR3 mutations on the growth plate [33,34]. In cases with SHOX deletion, the phenotype is highly variable; the moment of diagnosis may be from birth in Langer Mesomelic Dysplasia, children with SGA, or adult life when the diagnosis is established due to family gene analysis. The generalist, paediatrician, and endocrinologist may notice the short stature from childhood; still, the diagnosis of SHOX deletion is particularly challenging in preschool age when skeletal mesomelic shortening and Madelung deformity are not apparent [28]. Identifying patients with SHOX deficiency as early as possible is essential because GH treatment is most effective when started at a young age. There are no treatments provided for short stature caused by inactive mutations of the CNP receptor gene (*NPR2*) or the ACCAN gene mutation.

Considering the importance of an early diagnosis and treatment, special attention should be paid to the family history of short stature, growth parameters at birth, and the clinical and radiological examination. Our cases emphasize this aspect, especially in patients with a family history. The clinician must be aware of the genetic causes of idiopathic short stature to properly select patients for genetic testing and interpret the genetic tests in the clinical and therapeutic context.

## Figures and Tables

**Figure 1 diagnostics-13-00105-f001:**
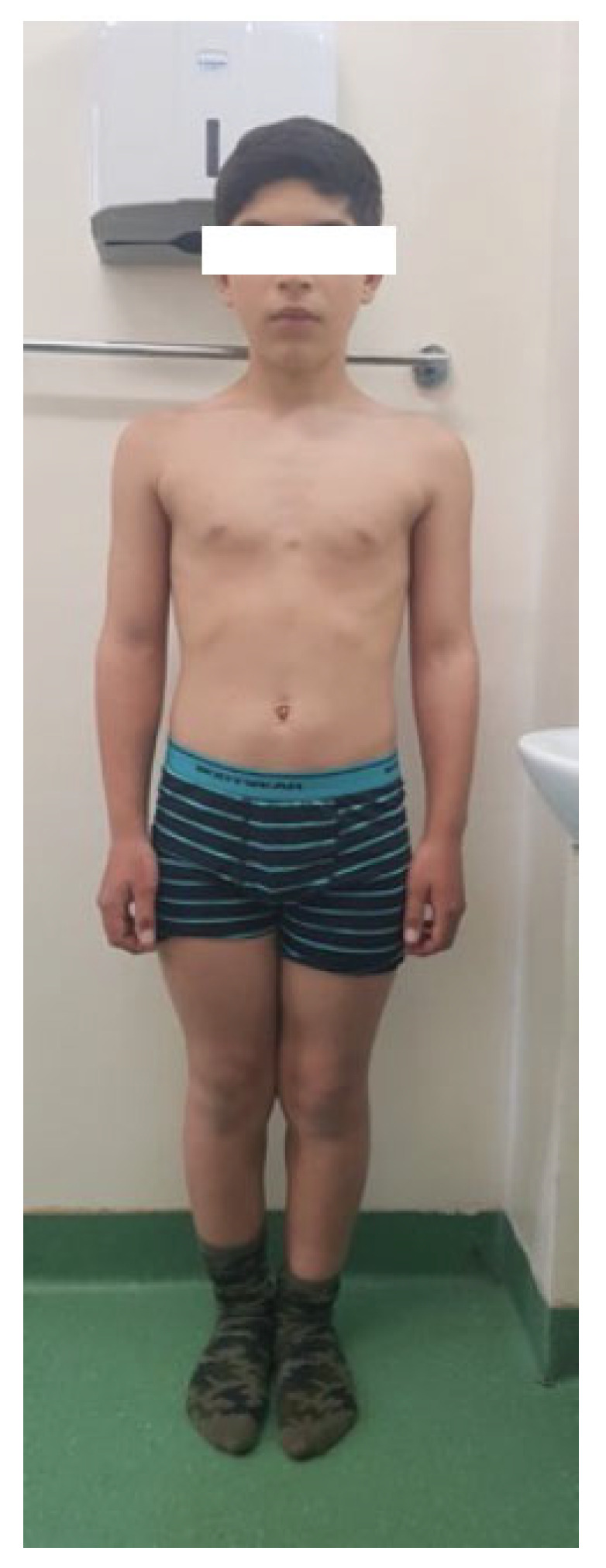
Case report 1 with mesomelic short stature: height −2.64 SD, BMI at 77th percentile, MPH −2.14 SD, extremities/trunk ratio ≤ 2.64, and sitting height/height >2.5 SD). The genetic test confirms the SHOX gene mutation.

**Figure 2 diagnostics-13-00105-f002:**
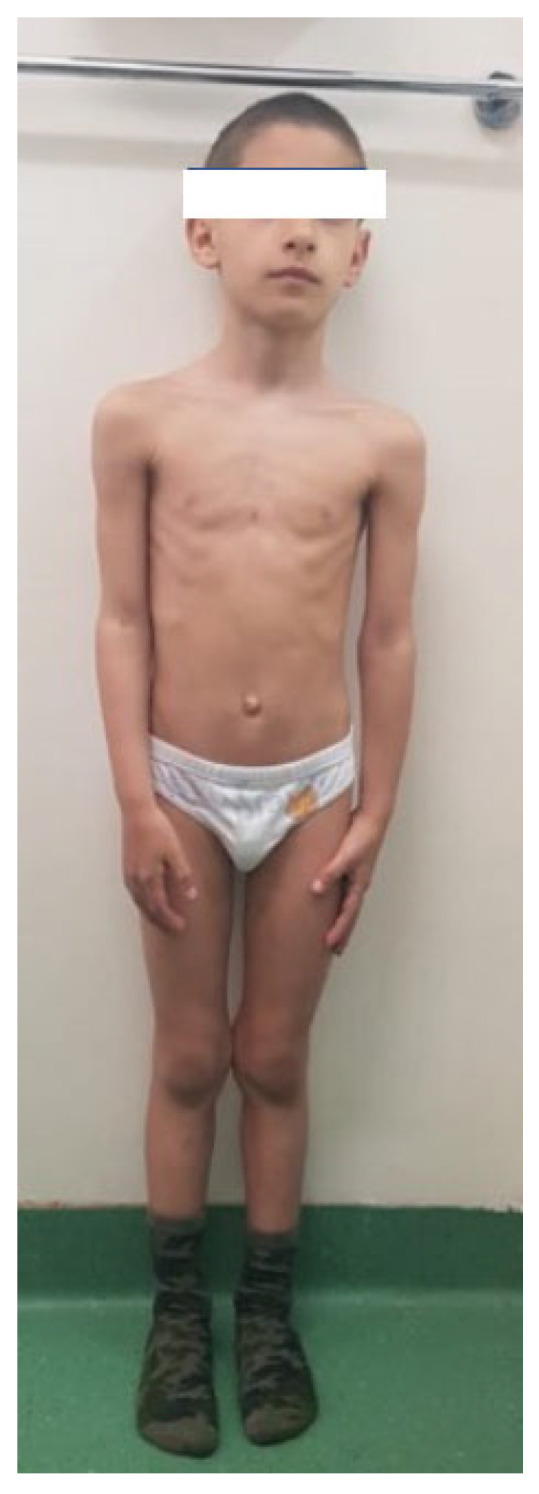
Case report 2: a nine-year-old boy born SGA. Treatment with rGH was started at the age of 5, with a height gain from −3.6 to −2.6 SD in 4 years. The clinical exam has excluded a mesomelic short stature.

**Figure 3 diagnostics-13-00105-f003:**
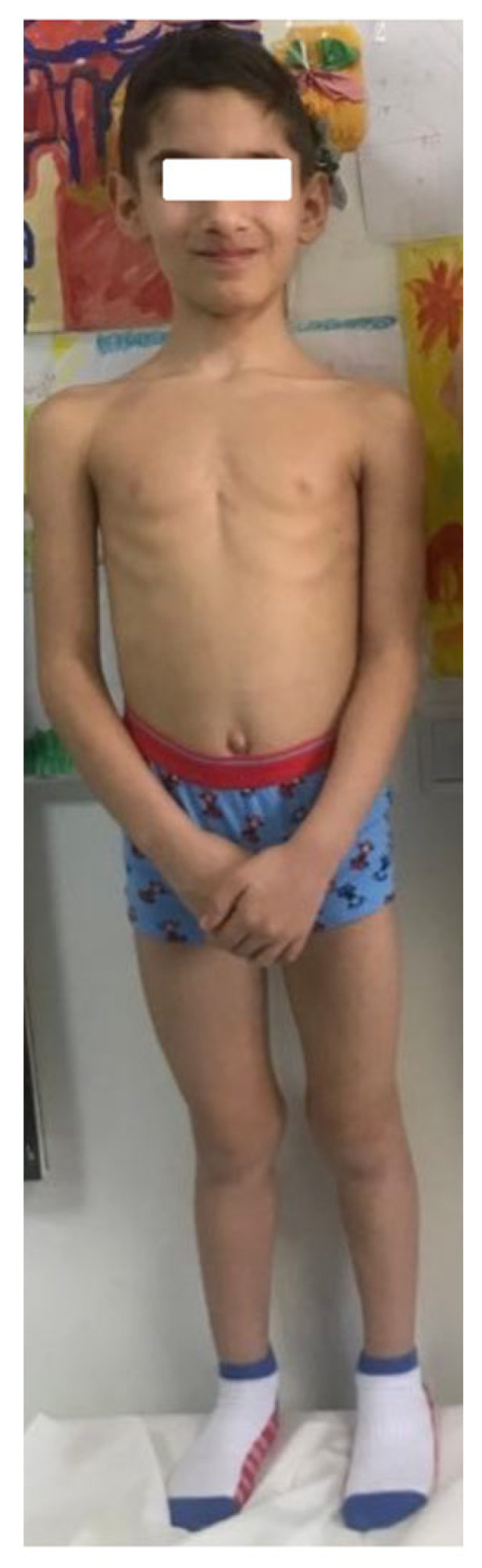
Case report three: eight-years-and-a-half-old boy, diagnosed at seven years with harmonious short stature: −3.2 SD, BMI 3rd percentile, and MPH −1.03 SD.

**Figure 4 diagnostics-13-00105-f004:**
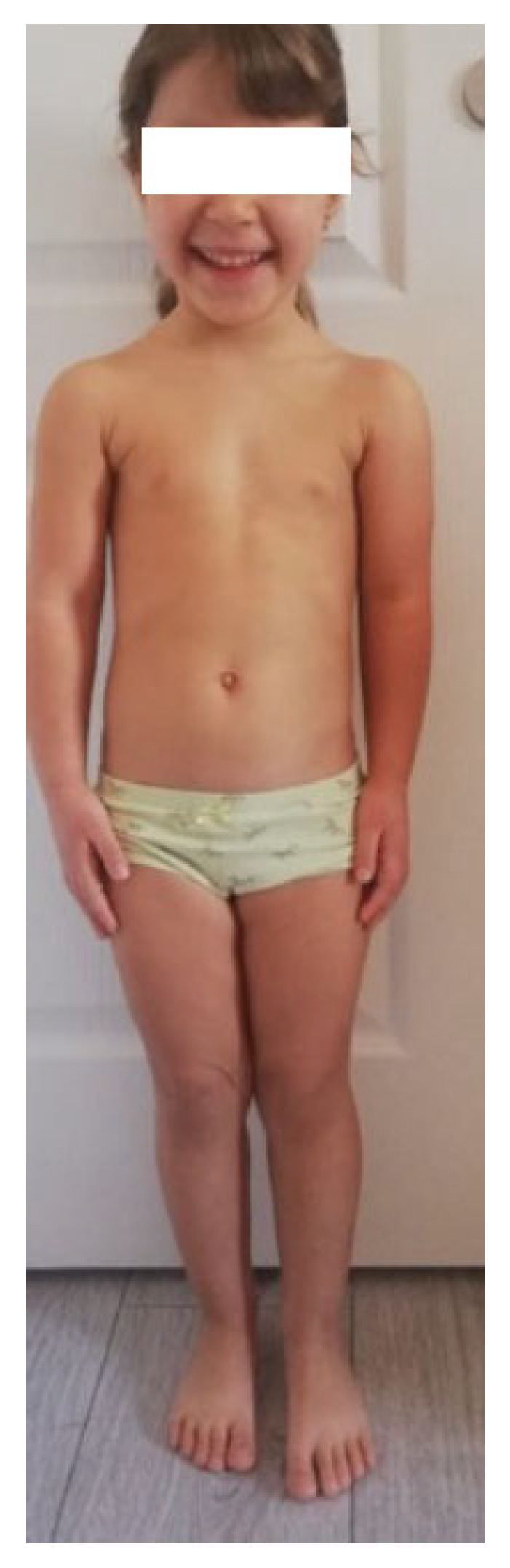
Case report 4: a five-year-old girl with low normal birth weight, height −1.6 SD, BMI 29th percentile, tested positive for *SHOX* deletion due to her family history. She is too young to evaluate a mesomelic shortening.

**Figure 5 diagnostics-13-00105-f005:**
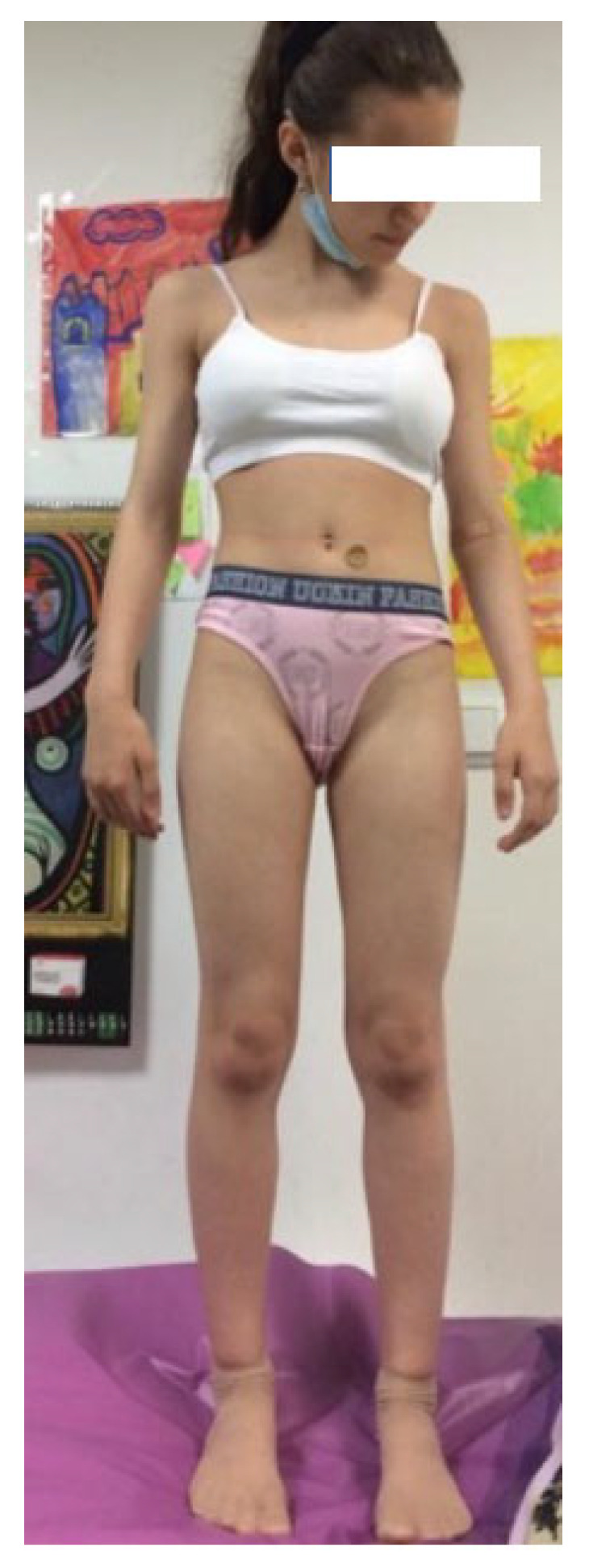
Case report 5: 11-year-old girl with height −1.41 SD, MPH −1.03 SD, and tested negative for *SHOX* deletion.

**Figure 6 diagnostics-13-00105-f006:**
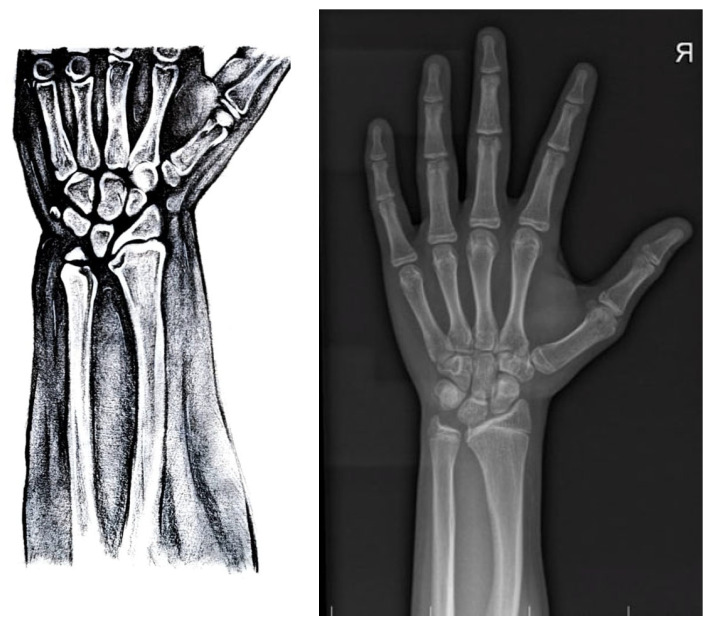
Radiological findings in SHOX deficiency (adapted from Vanelli et al. 2020).

**Figure 7 diagnostics-13-00105-f007:**
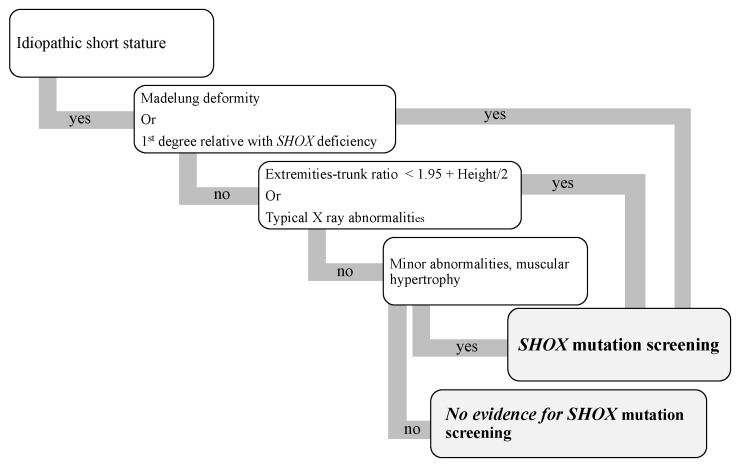
Diagnostic algorithm for SHOX mutation screening (adapted from Vanelli 2020, Binder 2011, 2018 and Fukami 2016).

## Data Availability

Data sharing is not applicable to this article.

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
