# Peer review of "SHOX Deletion and Idiopathic Short Stature: What Does the Clinician Need to Know? Case Series Report"

_diagnostics, 2022, doi:10.3390/diagnostics13010105_

Round 1
Reviewer 1 Report
Dear authors,
I have now completed the review of the manuscript titled "SHOX deletion and idiopathic short stature: what does the clinician need to know? Case series report."
In the present study, the authors presented kindred of five children evaluated for short stature or low normal stature, diagnosticated initially as idiopathic short stature, familial short stature, or small for gestational age.
The manuscript is interesting and, in general, fair written.
I have some suggestions to further improve the quality of the manuscript.
1. In lines 55-56, the references seem missing. Or is it about reference [2, 4, 5]? I would be glad if the authors clarify this.
2. I suggest authors clarify if they did further genetic sequencing (i.e., whole exome), and the result of the analysis.
3. Did the parent of the suggested cases also diagnose as SHOX haploinsufficiency? It would be better if the familial histories were explained.
4. What is the future scope of the proposed research, authors have described the limitations in a good way, and I suggest that these can be the future scope of the work.
5. The conclusions section is single-spaced.
Reviewer 2 Report
I was very pleased to see the submitted manuscript, which addresses 5 case reports demonstrating a co-relation between SHOX gene mutation and short stature. These studies will definitely enhance in addressing the understanding between the patients and the clinicians. The manuscript is straightforward and the case studies are convincing. However, I believe that a few improvements could increase the impact of this work. Especially, the case studies reported in this study is very low in number to make a significant observation and conclusion. Unless the disease prevalent is very rare, it is understandable that more case studies can be incorporated to convey a strong conclusion. These are my suggestions:
1. It is an interesting report presented by the authors. Please mention in the discussion about the percentage/ratio of the affected siblings when either both or a single parent carries SHOX gene mutation.
2. In the introduction, please mention how much of the world population is affected.
3. It is unclear what the authors used (source of the sample e.g. blood) to isolate the genomic DNA from these patients for genetic testing to analyze SHOX mutation. Please mention it in the methods section.
4. In case 5, please elaborate the cause of the short stature. Normal expression of the SHOX gene suggests that other factors could be involved. Did the authors analyze the growth hormone and other hormone parameters for this patient?
5. Please explain why case 5 was relevant to be included if the title of the manuscript is about the co-relation between SHOX mutation and short stature.
6. The authors have mentioned the cause of bone shortening include contribution of various factors such as extracellular matrix proteins, cytokines, hormones etc. Please explain, when the SHOX mutated patients are treated using recombinant-growth hormones, whether these mentioned factors are targeted that support bone lengthening process.
7. In the discussion please mention what could be an alternative to recombinant-growth hormone treatment when there may exist ethical concerns in certain parts of the world?
8. Figure 6, apart from the bone length, did the authors also observe a change in the bone diameter?
9. The authors have provided a schematic for SHOX mutation diagnosis in the affected patients. What is the earliest stage (months/years) when the short stature begins to be obvious?
